# Rose (*Rosa gallica*) Petal Extract Suppress Proliferation, Migration, and Invasion of Human Lung Adenocarcinoma A549 Cells through via the EGFR Signaling Pathway

**DOI:** 10.3390/molecules25215119

**Published:** 2020-11-04

**Authors:** Won-Chul Lim, Hyo-Kyung Choi, Kyung-Tack Kim, Tae-Gyu Lim

**Affiliations:** 1Korea Food Research Institute, Wanju-gun, Jeollabuk-do 55365, Korea; 07934@kfri.re.kr (W.-C.L.); chkyoung@kfri.re.kr (H.-K.C.); tack@kfri.re.kr (K.-T.K.); 2Department of Food Science & Biotechnology, Sejong University, Gwangjin-gu, Seoul 05006, Korea

**Keywords:** anticancer, invasion, migration, phytochemicals, proliferation, rose petal extract

## Abstract

We sought to investigate the effect of rose petal extract (RPE) on the proliferation, migration, and invasion of cancer cells. RPE significantly inhibited the growth of lung and colorectal cancer cell lines, with rapid suppression of A549 lung cancer cells at low concentrations. These effects occurred concomitantly with downregulation of the cell proliferation mediators PCNA, cyclin D1, and c-myc. In addition, RPE suppressed the migration and invasion of A549 cells by inhibiting the expression and activity of matrix metalloproteinase-2 and matrix metalloproteinase-9 (MMP-2 and -9). We hypothesize that the suppressive activity of RPE against lung cancer cell proliferation and early metastasis occurs via the EGFR-MAPK and mTOR-Akt signaling pathways. These early results highlight the significant potency of RPE, particularly for lung cancer cells, and warrant further investigation.

## 1. Introduction

Malignant cells invade and spread to distant parts of the body due to abnormal cell growth and behavior. According to statistics reported by the International Agency for Research on Cancer, although the number of lung cancer diagnoses in 20 countries accounted for approximately 11.6% of total cancer cases, the mortality rate was the highest at 18.4% [1]. Methods for treating cancer include surgery, radiation therapy, and chemotherapy. Depending on the stage of cancer progression, combination therapies can help prevent recurrence after successful surgery, or when surgery is impossible. Although chemotherapy has to date focused on synthetic chemicals, research into the development of anticancer drugs using natural compounds has attracted interest due to the potential for lower toxicity. Paclitaxel was developed from a plant of the genus Taxus and is a natural anticancer compound that suppresses cell division [2].

Cancer cells metastasize to distant sites through abnormal proliferation, migration, and invasion. Cyclin D1 and c-myc are well known oncogenes that are overexpressed in cancer cells, driving uncontrolled cell growth and tumor formation [3]. After tumorigenesis, primary tumors migrate to neighboring cells and invade the blood or lymphatic vessels by decomposing the extracellular matrix through enzymes such as matrix metalloproteinase (MMPs). Metastasis is achieved by invading cells gaining resistance to anoikis in the floating state, which then circulate and settle in distant organs [4]. Metastatic tumors are often fatal, with patients facing dismal five-year survival rates.

EGFRs are a large family of receptor tyrosine kinases overexpressed in many types of cancer, including breast, lung, esophagus, head, and neck cancer, and their abnormal activity is often responsible for the development, growth, and metastasis of tumor cells. For this reason, EGFR is a major marker for cancer diagnosis [5]. Overexpression of receptors such as EGFR, transforming growth factor beta (TGF_β_) receptor, c-met, and tropomyosin receptor kinase B (TrkB), and growth factors such as EGF, TGF, HGF, and BDNF have been found to drive aggressive and resistant cancer phenotypes. EGFR activation regulates the expression of proteins related to cancer cell growth, signaling, differentiation, adhesion, migration, and survival through the MAPK, Akt, and STAT signaling pathways. The MAPK pathway can continuously stimulate kinases such as Ras, Raf, and MEK [6].

Plant foods are becoming an important healthy diet that can provide vast components such as vitamins, minerals, and dietary fiber. Rose (*Rosa gallica*) is an ornamental plant used for aesthetic purposes in gardening and landscaping. Rose petals have been used in many countries as an ingredient in tea, jam, wine, and juice [7]. Its petals contain phenolic acid, flavonol, and anthocyanin, and are sometimes used as nutritional sources. In particular, rose petals have been reported to harbor various pharmacological activities such as insecticidal, anti-allergic, antioxidant, skin anti-inflammatory, and antibacterial effects [8,9,10,11,12]. However, research into the anticancer activity of rose petals has been limited. Here, we sought to investigate the effect of rose petal extract on the proliferation, migration, and invasion of lung and colon cancer cells.

## 2. Results

### 2.1. Rose Petal Extract Suppresses Lung and Colon Cancer Cell Proliferation

We assessed the effect of rose petal extract (RPE) on cancer cell proliferation (lung cancer cells: A549 and H1299; colon cancer cells: HCT116 and HT29) by measuring cell viability following 100, 200 and 400 μg/mL treatment with RPE after 24, 48 and 72 h. The proliferation of A549, H1299, HCT116 and HT29 cells was significantly reduced following exposure to RPE (Figure 1A–D). Notably, RPE showed stronger potency against lung cancer cells than the colon cancer cells. Whereas 100 μg/mL of RPE did not fully suppress colon cancer cell proliferation, the proliferation of A549 and H1299 was significantly reduced. However, the proliferation of both lung and colon cancer cells was effectively abolished by 200 and 400 μg/mL of RPE. Interestingly, in normal lung Mlg and WI-38 cells, a proliferation rate of 80% or more was observed at all treatment concentrations of RPE, and thus, the growth of lung normal cells was not affected (Appendix A).

### 2.2. Rose Petal Extract Reduces Cyclin D1 and c-myc Expression and Enhances PTEN Expression in A549 Cells

We next analyzed cyclin D1 and c-myc expression levels in A549 and HCT116 cells to examine whether RPE affects the expression of proteins associated with cancer cell proliferation [13]. Similar to the effect of RPE on cancer cell proliferation, the expression of cyclin D1 and c-myc expression was dose-dependently reduced by RPE in A549 cells (Figure 2A,B) and HCT116 (Figure 2C,D). Next, we evaluated PTEN expression after RPE treatment, as it has been reported that PTEN negatively regulates cyclin D1 and c-myc expression [14]. Interestingly, the expression levels of PTEN were increased by RPE treatment in A549 and HCT116 cells in a dose-dependent manner (A549, Figure 2A,B; HCT116, Figure 2C,D). We hypothesize that RPE-induced PTEN upregulation might be an upstream mechanism involved in cyclin D1 and c-myc expression. Next, we confirmed the anti-proliferative activity of RPE by assessing PCNA expression in the nucleus of A549 cells, in which expression of cell proliferation-related factors were altered more by RPE (Figure 2E). In support of the cell proliferation results (Figure 1), PCNA expression in the nucleus dramatically reduced with treatment of 100 μg/mL RPE.

### 2.3. Rose Petal Extract Inhibits Cell Migration and Invasion by A549 Cells

We investigated changes in cell migration and invasion following RPE treatment in A549 cells and detected dose-dependent inhibition of migration following RPE exposure (Figure 3A). Subsequently, we analyzed the effect of RPE on invasiveness of A549 cells using a trans-well assay system. It was observed that A549 cells possess vigorous invasive capabilities, and RPE effectively diminished this phenotype in a dose-dependent manner (Figure 3B). Next, we evaluated the effect of RPE on the expression and activity of MMP-2 and -9 in A549 cells. A dramatic reduction in MMP-2 and -9 activity was observed using a zymography assay (Figure 3C). The expression of MMP-2 and -9 was also reduced by RPE treatment in A549 cells (Figure 3D). In addition, cancer cells proliferate and metastasize after they migrate and invade and settle in distant secondary site. When A549 cells exposed to RPE were cultured in a new cell culture plate, reattachment did not occur compared to untreated cells (Appendix A). Taken together, we hypothesize that RPE has anti-metastatic activity by suppressing cancer cell migration and invasion, which is associated with lower activity and expression of MMP-2 and -9.

### 2.4. Rose Petal Extract Regulates EGFR-c-Raf-MEK and the mTOR-Akt Signaling Pathway in A549 Cells

Previous research has shown that aberrant activation of EGFR is closely associated with cancer cell properties such as growth, malignancy, and metastasis [15,16,17]. Based on this background, after activation of EGFR by EGF, a ligand of EGFR, changes in proliferation marker expression of the A549 cell line were measured. As a result, activation of EGFR by EGF significantly increased the expression of cyclin D1 and c-myc. However, RPE suppressed the increased proliferation by EGF in lung cancer A549 cells (Appendix A). Subsequently, we sought to evaluate the effect of RPE on the phosphorylation of the EGFR-c-Raf-MEK pathway in A549 cells. Treatment with RPE alleviated the phosphorylation of these proteins (Figure 4A,B), and we next assessed the effect of RPE on the mTOR-related signaling pathway. mTOR and Akt phosphorylation was decreased by RPE treatment in A549 cells (Figure 4C,D). These results indicate that RPE attenuates the mTOR-Akt pathway via upregulation of the negative regulator of mTOR, PTEN. In particular, RPE showed higher inhibitory activity against phosphorylation of EGFR than that of mTOR. Therefore, after co-treatment with erlotinib and RPE, known as EGFR inhibitor, respectively, the suppressive actions of proliferation, migration, and invasion of the A549 cells were confirmed. As a result, RPE suppressed similarly to erlotinib for proliferation, migration, and invasion of cancer cells, and the inhibitory activity was further increased by co-treatment of erlotinib and RPE (Appendix A and Figure 4E–H). The EGFR-c-Raf-MEK and PTEN-mTOR-Akt signaling pathways are closely associated with cancer cell properties, and we therefore suggest that the modulation of EGFR-c-Raf-MEK and PTEN-mTOR-Akt by RPE could be the underlying mechanism responsible for RPE’s anti-cancer activity.

## 3. Discussion

Abnormal cell division is a hallmark characteristic of cancer and can occur due to unchecked DNA damage, which is distinct from normal cells in which programmed cell death occurs [18,19,20]. Loss of control of cell cycle checkpoints by cancer cells can contribute to unrestrained cell proliferation, metastasis, and invasion of secondary tumor sites [21]. Chemotherapeutic agents can inhibit malignant tumor growth by targeting cell division and are used alone as well as in combination with surgery and radiotherapy. The primary mechanisms through which such agents inhibit cancer cell proliferation include the induction of apoptosis, and the suppression of cell division, angiogenesis, invasion, and metastasis. In this study, we investigated the anticancer effects of RPE on cancer cell proliferation, migration, and invasion. RPE showed inhibitory activity against proliferation of lung cancer (A549 and H1299) and colon cancer cell lines (HCT116 and HT29) as determined by CCK-8 assay. Changes in the expression of PCNA, a cofactor of DNA polymerase, were confirmed by immunostaining in the A549 cell line, which was the most susceptible to RPE treatment in terms of proliferation inhibition. PCNA is a marker capable of informing the diagnosis and prognosis of various cancers due to its high expression in proliferating cells. In the current study, we found that even the lowest concentration of RPE (100 μg/mL) was effective in suppressing the proliferation of A549 cells by down-regulating PCNA expression. In addition, RPE significantly reduced the expression of the oncogenes cyclin D1 and c-myc, which are closely associated with proliferation, while significantly increasing the expression of the oncogene repressor PTEN in both A549 lung cancer cells and HCT116 colorectal cancer cells. Cyclin D1 is used as a diagnostic marker as it is a commonly overexpressed oncogene in various carcinomas. Similar to cyclin D1, the c-myc oncogene is known to play a major role in the development of cancer and is dysregulated in most malignancies, including lung cancer. In particular, these tumor genes contribute to the formation of more aggressive and invasive tumors through interplay with each other [22]. PTEN is a tumor suppressor gene that is frequently mutated in cancer. Studies have shown that a decrease in PTEN levels causes cell overgrowth with potential cancer progression [23]. In the context of cancer treatment, it is important to inhibit cancer metastasis as well as suppress cancer growth. More than 90% of cancer deaths are associated with metastasis away from the primary tumor, and metastasis represents a significant challenge in cancer treatment [24]. During the metastatic process, cancer cells migrate to adjacent cells and invade them through the blood or lymphatic vessels [5]. As shown in the results of the wound healing and Matrigel invasion assays, the motility and invasion of A549 lung cancer cells was reduced by more than half by RPE concentrations up to 400 μg/mL. MMP is a metal-containing protease, which is known to play a major role in cancer cell invasion and metastasis. Of the several types of MMPs classified according to substrate preference, type IV collagenase is divided into gelatinase A (MMP-2) and gelatinase B (MMP-9), and these enzymes break down the extracellular matrix, which consists of gelatin, collagen, laminin, proteoglycan, and fibronectin [25]. RPE, which showed significant inhibitory activity against cell migration and invasion, likewise inhibited both the expression and activity of MMP-2 and MMP-9. Of particular note, RPE completely eliminated the activity of MMP-2 and MMP-9 at a concentration of 100 μg/mL. From the results of this zymographic and immunoblotting analysis, it was hypothesized that RPE inhibits migration and invasion (the intermediate stage of metastasis) by suppressing the activity of MMP-2 and MMP-9. On the other hand, cancer cells that have successfully invaded blood or lymphatic vessels are resistant to anoikis and metastasize to distant sites. In this study, it was confirmed through colony formation assays [26] that RPE interferes with the survival of cancer cells in anchorage-independent conditions (Appendix A). Therefore, RPE is thought to inhibit the unique properties of cancer cells that are essential for invading tumor cells to metastasize to secondary regions.

The upstream pathways that regulate the cell cycle include the PI3K/Akt, NF-κB, and ERK1/2 pathways, among which the PI3K/Akt pathway is of pivotal importance. When Akt phosphorylation is inhibited, enzymes of the cyclin-dependent kinase (CDK) family (which are downstream signaling intermediates of the cell cycle) are inhibited in a process regulated by cyclin D1 [23,27]. According to results from previously reported studies, natural products such as epigallocatechin, luteolin, and kaempferol exhibit anti-proliferative effects through inhibition of the PI3K/Akt pathway and upregulation of PTEN, and anti-metastatic effects potentiated by MMP inhibition [28,29,30].

Evidence suggests that the overexpression of various receptors and growth factors can drive the development of aggressive and resistant cancer phenotypes. EGFR refers to a family of receptor tyrosine kinases overexpressed by many types of cancer, including breast, lung, and colon cancer. EGFR is a key regulator of complex signaling cascades related to cancer cell growth, signaling, differentiation, adhesion, migration, and survival. Due to its complex role in the progression of cancer, EGFR is an important target for cancer therapy [31]. The activation of EGFR plays an important role in the development and growth of tumor cells and is particularly involved in cellular responses relating to proliferation and apoptosis [32]. EGFR activation is triggered by binding between the EGF receptor and the EGF ligand, resulting in the phosphorylation of specific tyrosine residues at the receptor C-terminus and the induction of receptor dimerization [6]. Activated receptors stimulate the c-Raf-MEK pathway and PTEN-mTOR-Akt pathway, resulting in cell proliferation, migration, survival, and adhesion [33,34]. In this study, RPE not only inhibited cyclin D1 and c-myc expression under basal conditions for A549 cells, but also suppressed cancer growth under artificial EGF-treated conditions (Appendix A). It was found that the proliferative inhibitory activity of RPE was associated with suppression of the EGFR signaling pathway. Moreover, higher EGFR inhibitory activity was observed for RPE than for erlotinib, a clinical EGFR inhibitor (Appendix A).

## 4. Materials and Methods

### 4.1. Reagents

*Rosa gallica* petals grown in Turkey were obtained through GN Bio (Gyeonggi, Korea). Lung cancer (A549 and H1299) and colon cancer (HCT116 and HT29) cell lines were purchased from Korean Cell Line Bank (Seoul, Korea). Dulbecco’s modified Eagle’s medium (DMEM), fetal bovine serum (FBS), and penicillin/streptomycin solution were purchased from GIBCO^®^ Invitrogen (Auckland, New Zealand). Additionally, 3-(4,5-dimethylthiazol-2-yl)-5-(3-carboxymethoxyphenyl)-2- (4-sulfophenyl)-2H-tetrazolium (MTS) solution was purchased from Promega (Madison, WI, USA). Primary antibodies for p-EGFR^Tyr1068^, EGFR, p-c-Raf, c-Raf, PTEN, MMP-9, -2, p-MEK1/2, p-mTOR, m-TOR, p-Akt^Ser473^, and Akt were obtained from Cell Signaling Biotechnology (Danvers, MA, USA). Antibodies against cyclin D1, c-Myc, MEK1/2 and β-Actin were purchased from Santa Cruz Biotechnology (Santa Cruz, CA, USA).

### 4.2. Preparation of Rose Petal Extract (RPE)

Rose petals were ground in a blender to obtain a fine powder. Ten grams of the dried powder was mixed with 1000 mL of 70% (*v*/*v*) ethanol and extracted at 70 °C for 3 h using a reflux condenser. The extract was then filtered through No.2 filter paper (Whatman, Maidstone, UK). The solvent was subsequently evaporated, and the product was freeze-dried.

### 4.3. Cell Culture

Mlg, WI-38 and HCT116 cells were cultured in DMEM, while the A549, H1299 and HT29 cells were cultured in RPMI. The medium for cell culture was supplemented with 10% (*v*/*v*) FBS and antibiotics (100 μg/mL of streptomycin and 100 U/mL of penicillin) at 37 °C in a humidified 5% CO_2_ incubator.

### 4.4. Cell Proliferation Assay

Lung cancer (A549 and H1299) and colon cancer (HCT116 and HT29) cells were seeded (2 × 10^3^ cells/well) in 96-well plates and incubated for 12 h, before RPE was treated to the cells for 24, 48 or 72 h. Cell proliferation was measured by adding 10 μL/well of CCK-8 reagent (EZ-cytox; Dogen Bio, Seoul, Korea). After incubation with CCK-8 solution for 1 h at 37 °C, the absorbance was measured at 450 nm.

### 4.5. Immunofluorescence Assay

A549 cancer cells were seeded in coverslip plates (3 × 10^4^ cells/well) and incubated for 12 h. The cells were treated with RPE at concentrations of 0, 100, 200, and 400 μg/mL for 24 h. After washing with PBS, the cells were fixed with 4% formalin, permeabilized with 0.01% Triton X-100, blocked with 3% BSA, and incubated with anti-PCNA primary antibody at 4 °C for 12 h. After incubation, the cells were washed with PBS three times and incubated with anti-rabbit FITC-conjugated secondary antibody at rt for 1 h in the dark. After incubation, the cells were washed with PBS three times and counterstained with DAPI for 5 min. PCNA-positive cells were observed with a fluorescence microscope.

### 4.6. Western Blot Analysis

Protein samples from A549 and HCT116 cells were collected using 1 × cell lysis buffer (Cell Signaling, Danvers, MA, USA). The protein concentrations were analyzed with a Pierce^TM^ BCA Protein Assay Kit (Thermo Fisher Scientific, San José, CA, USA). The proteins were separated on 10% SDS-polyacrylamide gels (Bio-Rad, Hercules, California, USA) based on molecular weight for electrophoresis and then transferred to Immobilon P membranes (Millipore, Billerica, MA, USA). The membranes were blocked with 5% fat-free milk for 1 h and incubated with the specific primary antibodies at 4 °C overnight. After hybridization with an HRP-conjugated secondary antibody (Cell Signaling Technology, Danvers, MA, USA), the protein bands were visualized using ECL reagents (Bio-Rad) under a chemiluminescence imaging system (Uvitec Cambridge, UK).

### 4.7. Migration Assay

Cell migration was evaluated with a wound healing assay. The assay was carried out as described in previous studies [33,35,36] with few modifications. Briefly, the cells were seeded into 6-well plates at a density of 5 × 10^5^ cells/well and cultured until 90% confluence before the well surface was artificially scratched using a 200 μL pipette tip. The cells were washed twice with PBS to remove debris and detached cells before the cells were exposed to RPE at various concentrations. Photographic images were taken of each of the parallel lines and the Image J program was used for quantification of the results.

### 4.8. Invasion Assay

The invasiveness of the cancer cells was evaluated with a Matrigel-coated trans-well chamber. The cells were detached with trypsin/EDTA buffer and added to the chamber with serum-free medium containing RPE. The complete medium was then added to the lower chamber and the cells were incubated at 37 °C for 48 h. The non-invasive cells of the upper chamber were removed by cotton swabs and the invasive cells under the membrane were stained with cell staining solution. The invasive cells were photographed, and the Image J program was used to quantify the results.

### 4.9. Gelatin Zymography Assay

The activation of MMP-2 and -9 was evaluated using gelatin zymography as described in a previous study [37]. In brief, the media was collected after 48 h of treatment with RPE in A549 cells. Protein concentrations were measured using a Pierce^TM^ BCA Protein Assay Kit (Thermo Fisher Scientific, San José, CA, USA). The same amount of medium was loaded onto SDS-PAGE gels containing gelatin, and the gel was washed with renaturing buffer for 30 min at RT. For development, the gels were incubated for 24 h at RT and MMP-2 and -9 activity was visualized by staining the gels with 0.5% Coomassie brilliant blue.

### 4.10. Statistical Analysis

All statistical analyses were performed using SPSS software version 20.0 (SPSS Inc., Chicago, IL, USA). The data were expressed as mean values ± standard deviation (SD). Statistical significance was determined using Student′s *t*-test for single statistical comparisons and *p*-values of <0.05 were regarded as significantly different.

## 5. Conclusions

Our findings highlight the inhibitory effect of RPE against the hallmarks of lung cancer cells including proliferation, migration, and invasion, which was associated with the inhibition of EGFR activation and subsequent c-Raf-MEK and PTEN-mTOR-Akt signaling. To clinically adapt current research, more studies, including clinical trials, are necessary.

## Figures and Tables

**Figure 1 molecules-25-05119-f001:**
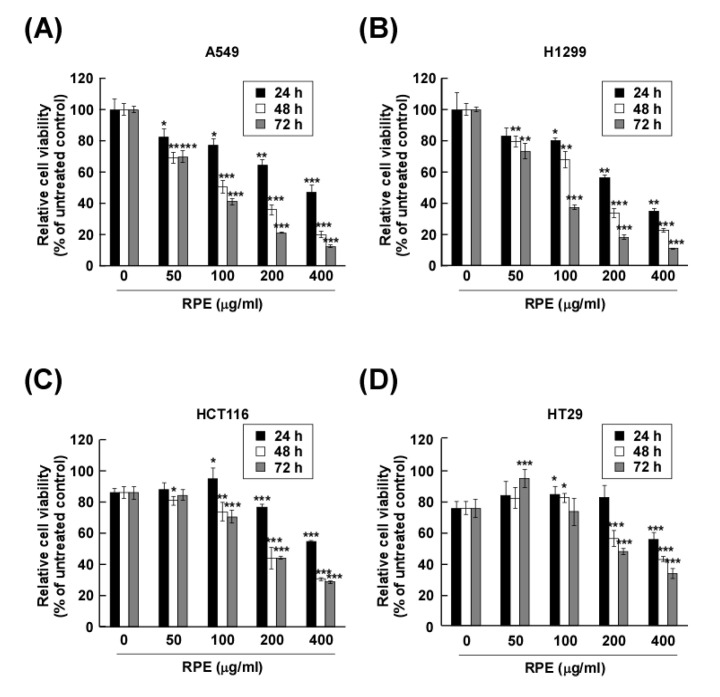
Effect of rose petal extract (RPE) on proliferation of human lung and colon cancer cells. (**A**,**B**) A549 and H1299 cells and (**C**,**D**) HCT116 and HT29 cells were treated with RPE at a concentration of 0, 100, 200, or 400 μg/mL for 24, 48, or 72 h, and then a CCK-8 assay was performed. Relative cell viability is represented as means ± standard deviation (*n* = 3) as * *p* < 0.05, ** *p* < 0.01, *** *p* < 0.005, compared with untreated group.

**Figure 2 molecules-25-05119-f002:**
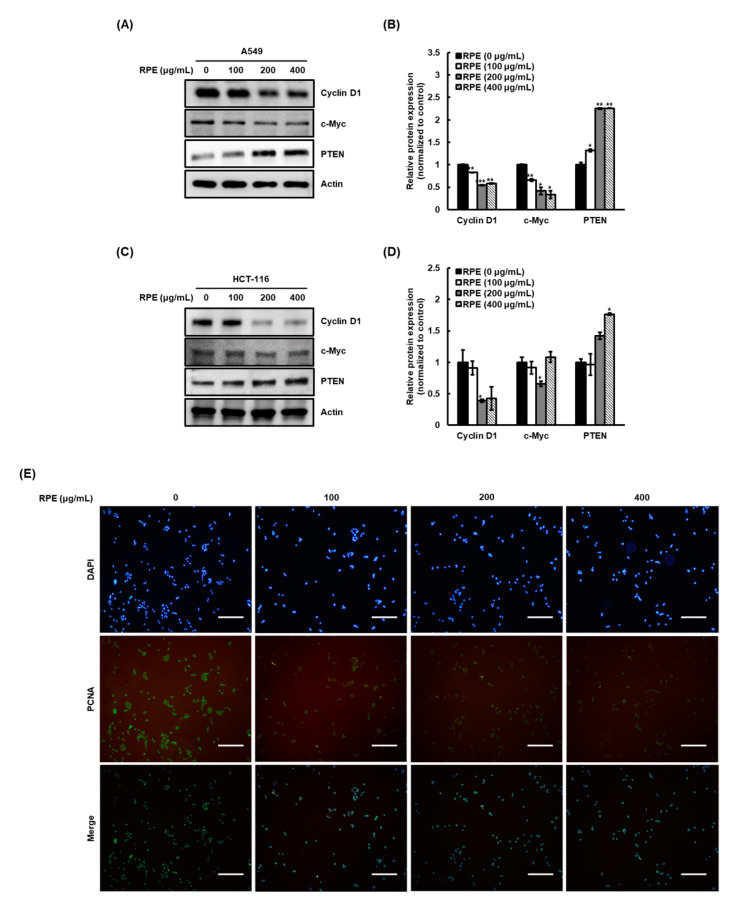
Effect of RPE on expression of proliferation-related protein in A549 and HCT116 cells. These cells were treated with RPE at a concentration of 0, 100, 200, or 400 μg/mL for 24 h. The expression of cyclin D1, c-myc, and PTEN of (**A**,**B**) A549 and (**C**,**D**) HCT116 cells were detected by Western blotting and densitometric analysis. β-actin was used as loading control. Relative expression is represented as means ± standard deviation (*n* = 3) as * *p* < 0.05, ** *p* < 0.01, *** *p* < 0.005, compared with untreated group. (**E**) The expression of PCNA of A549 cells was observed by immunofluorescence staining, and then representative images were indicated. Size of the scale bar in the images refer to 100 μm.

**Figure 3 molecules-25-05119-f003:**
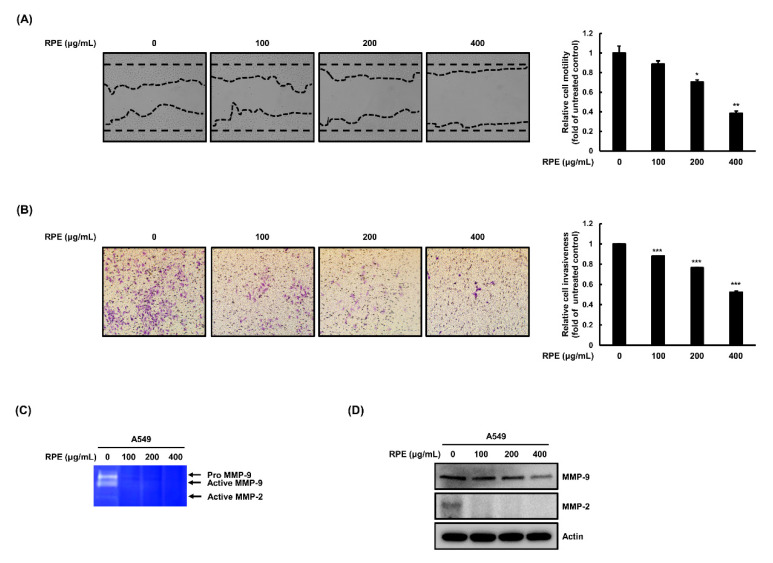
RPE reduces migration and invasion of human adenocarcinoma A549 cells. (**A**) Scratched A549 cells were treated with RPE for 24 h, and then the wound healing area was photographed and measured with the Image J program. (**B**) The cells that invaded through the extracellular matrix-like membrane were measured with the Matrigel invasion assay. Relative motility or invasiveness of cells is represented as means ± standard deviation (*n* = 3) as * *p* < 0.05, ** *p* < 0.01, *** *p* < 0.005, compared with untreated group. Size of the scale bar in the images refer to 100 μm. (**C**) Activities of matrix metalloproteinase-2 (MMP-2) and matrix metalloproteinase-9 (MMP-9) were detected by zymography and (**D**) expression of those were measured with Western blotting analysis. Representative bands are indicated.

**Figure 4 molecules-25-05119-f004:**
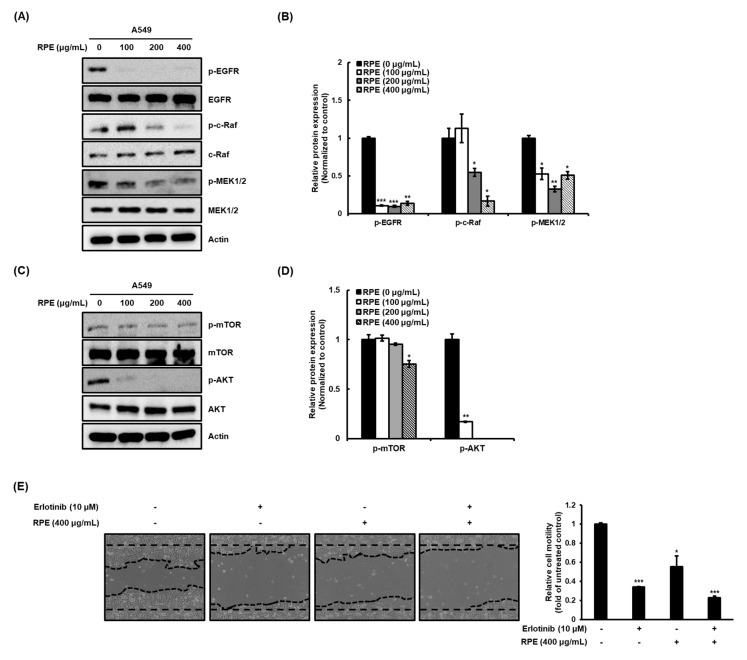
RPE suppresses proliferation, migration, and invasion by EGFR-c-Raf-MEK and mTOR-Akt signaling pathways. (**A**) Scratched A549 cells were treated with RPE for 24 h, and then the wound healing area was photographed and measured with the Image J program. (**B**) The cells that invaded through the extracellular matrix-like membrane were measured with the Matrigel invasion assay. (**C**) Activities of MMP-2 and MMP-9 were detected by zymography and (**D**) expression of those were measured with Western blotting analysis. Representative bands are indicated. (**E**) Wounded A549 cells were treated with erlotinib or RPE for 24 h, and then the migrated area was photographed and measured with the Image J program. (**F**) The invasive cells were measured with the Matrigel invasion assay. Size of the scale bar in the images refer to 100 μm. (**G**) The expression of p-EGFR, EGFR, and MMP-9 was determined by Western blotting and (**H**) densitometric analysis. β-actin was used as loading control. Relative motility, invasiveness or expression are represented as means ± standard deviation (*n* = 3) as * *p* < 0.05, ** *p* < 0.01, *** *p* < 0.005, compared with untreated group.

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
