# Peer review of "Rose (Rosa gallica) Petal Extract Suppress Proliferation, Migration, and Invasion of Human Lung Adenocarcinoma A549 Cells through via the EGFR Signaling Pathway"

_molecules, 2020, doi:10.3390/molecules25215119_

Round 1
Reviewer 1 Report
Major comments:
- Although I appreciate very much the effort of the authors who extensively evaluated the biological effects of RPE on cancer cells, I don't see a very promising conclusion come out from the findings. For example, do EGFR-MAPK and mTOR-Akt signaling pathways work in parallel or in sequence under the treatment of RPE?
- Related to 1, How did RPE inhibit EGFR? Same issue can be applied to p-mTOR. It is great to see that RPE can inhibit so many key factors involved in cancer proliferation and migration, but how did they get inhibited by RPE?
Minor comments:
- In Fig. 1, the "Mlg" cells are mouse, but not human, lung fibroblasts. In fact, fibroblasts tend to be more resilient to drug treatment than epithelial cells. So Mlg cells are not the relevant control normal cells. Furthermore, the statistics in Fig. 1 and 4 should be considered multiple comparisons, so student's t-test may not be suitable.
- In Fig. 2, c-myc downregulation by RPE is not convincing, especially in HCT116. In addition, although it is interesting to see upregulation of Pten by RPE, further confirmation on the level of its substrate PIP3 should help to solidify this result. PCNA should be more convincingly detected by Western blot analysis.
- In Fig.3, while MMP9 expression in the cells is still abundant at various doses of drug treatment, both pro and active MMP9 are invisible in the culture media. Does it mean RPE can inhibit MMP9 secretion?
- In Fig. 4C, p-mTOR is not inhibited by RPE in A549. How did p-Akt get downregulated under such circumstance? Furthermore, to prove that RPE inhibits cancer cell migration through negatively regulating EGFR activity, one should use EGFR agonist (e.g. EGF) but not inhibitor to firmly demonstrate this biological effect.
- To say that RPE has suppressive effect on cancer metastasis is just too premature based on current results.
- The legends for Fig. 4G and H are missing.
- The introduction section should include more literature review about RPE, bit not general information about cancer.
Author Response
Reviewer: 1
Major comments:
Comment. Although I appreciate very much the effort of the authors who extensively evaluated the biological effects of RPE on cancer cells, I don't see a very promising conclusion come out from the findings. For example, do EGFR-MAPK and mTOR-Akt signaling pathways work in parallel or in sequence under the treatment of RPE?
Response. We appreciate for the reviewer’s indication. The scientific novelties of current research are to find natural bioactive compound which can inhibit lung cancer and to suggest its underlying mechanism for the anti-cancer activity. We sought RPE reveals anti-proliferation activity on lung cancer cell by suppressing EGFR-MAPK and mTOR-Akt signaling pathway. Based on previous studies, EGFR is regulate c-Raf-MEK-ERK and mTOR-Akt signaling pathway as a upstream modulator in carcinogenesis [1,2].
Comment. Related to 1, How did RPE inhibit EGFR? Same issue can be applied to p-mTOR. It is great to see that RPE can inhibit so many key factors involved in cancer proliferation and migration, but how did they get inhibited by RPE?
Response. We appreciate for the reviewer’s comment. Previous papers have shown that natural phytochemical can act as a kinase inhibitor by direct binding on enzyme’s structure. Our research group already reported that butein, a natural chalcone chemical can directly inhibit EGFR kinase activity [3]. Based on our unpublished data, RPE significantly reduced recombinant EGFR(h) kinase activity. Thus, we assumed that the compound(s) in RPE might directly bind to EGFR for suppression the activity.
Minor comments:
Comment. In Fig. 1, the "Mlg" cells are mouse, but not human, lung fibroblasts. In fact, fibroblasts tend to be more resilient to drug treatment than epithelial cells. So Mlg cells are not the relevant control normal cells. Furthermore, the statistics in Fig. 1 and 4 should be considered multiple comparisons, so student's t-test may not be suitable.
Response. We appreciate for the reviewer’s comment. In this study the results of RPE, which showed a strong inhibitory effect on the proliferation of cancer cells, did not affect the proliferation of normal lung cells Mlg, are considered meaningful. However, as your comment, Mlg cells were different from the cancer cell species used in this study. Thus, in order to exclude differences in the effect of RPE by species, it was further analyzed whether RPE in normal cells of human species exhibits cytotoxicity. As expected, it was found that RPE did not affect the proliferation of WI-38 lung normal cells. The results of RPE cell proliferation in WI-38 cells were added to Figure S1B as shown in bellows. In addition, in all the results, the experimental groups were compared by setting the untreated group as a control group. It can be seen through statistical analysis that RPE or co-treatment with erlotinib and RPE significantly inhibited cancer cell growth compared to the control group.
Comment. In Fig. 2, c-myc downregulation by RPE is not convincing, especially in HCT116. In addition, although it is interesting to see upregulation of Pten by RPE, further confirmation on the level of its substrate PIP3 should help to solidify this result. PCNA should be more convincingly detected by Western blot analysis.
Response. We appreciate for the reviewer’s comment. We confirmed that RPE inhibited the proliferation of all four cell lines, but the expression of proliferation-related markers such as cyclin D1 and c-Myc, which I targeted, was most markedly changed in the A549 cell line. Therefore, the title of this study was suggested as the inhibitory effect of RPE on the proliferation of lung cancer A549 cells, not the inhibitory activity of RPE on the proliferation of lung and colon cancer. On the other hand, the reduction of mTOR and Akt phosphorylation was indicated as evidence that can support the result of PTEN increased by RPE. As suggested in the text, RPE is described as suppressing the mTOR-Akt pathway through upregulation of PTEN, a negative regulator of mTOR. The results of PCNA fluorescence immunostaining are presented as additional supportable results in addition to cyclin D1. In fact, it is judged that only the result of the expression of cyclin D1 reduced by RPE is sufficient, and even in the result of immunofluorescence, it can be confirmed that the expression of PCNA is somewhat reduced by RPE.
Comment. In Fig.3, while MMP9 expression in the cells is still abundant at various doses of drug treatment, both pro and active MMP9 are invisible in the culture media. Does it mean RPE can inhibit MMP9 secretion?
Response. We appreciate for the reviewer’s comment. Zymogram of MMP-9, Figure 3C revealed the activity of MMP-2 and -9. As the reviewer’s comment, RPE might reduce MMP-9 secretion. However, the endogenous MMP-9 expression is also attenuated by RPE in Figure 3D. Thus, we concluded RPE suppress both expression (may also secretion) and activity of MMP-2 and -9. (Line 195-196)
Comment. In Fig. 4C, p-mTOR is not inhibited by RPE in A549. How did p-Akt get downregulated under such circumstance? Furthermore, to prove that RPE inhibits cancer cell migration through negatively regulating EGFR activity, one should use EGFR agonist (e.g. EGF) but not inhibitor to firmly demonstrate this biological effect.
Response. We appreciate for the reviewer’s comment. The band in Figure 3C is the result with more than 3 times experiment. Based on multiple experimental results, the band intensity was analyzed using Image J program. As seen in Figure 3D, p-mTOR was statistically reduced by RPE treatment. As per the reviewer’s indication, to prove the EGFR-related biological activity of RPE it is necessary to using EGFR signaling activation system. Thus, we confirmed the effect of RPE using A549 cells which model is constitutive EGFR activation model [4,5].
Comment. To say that RPE has suppressive effect on cancer metastasis is just too premature based on current results.
Response. We appreciate for the reviewer’s indication. As per the reviewer’s indication, in vivo or clinical study is required to demonstrate clear anti-cancer metastatic activity of RPE. Indeed, pulmonary metastasis assay is now conducting in our lab for further study.
Comment. The legends for Fig. 4G and H are missing.
Response. We appreciate for the reviewer’s indication. We added the Figure legend to revised manuscript.
Comment. The introduction section should include more literature review about RPE, bit not general information about cancer.
Response. We appreciate for the reviewer’s suggestion. We inserted additional explanation about RPE in introduction section as below.
“Pant foods are becoming an important healthy diet that can provide vast components such as vitamins, minerals, and dietary fiber. Rose (Rosa gallica) is an ornamental plant used for aesthetic purposes in gardening and landscaping. Rose petals have been used in many countries as an ingredient in tea, jam, wine, and juice [7]. Its petals contain phenolic acid, flavonol, and anthocyanin, and are sometimes used as nutritional sources. In particular, rose petals have been reported to harbor various pharmacological activities such as insecticidal, anti-allergic, antioxidant, skin anti-inflammatory, and antibacterial effects [8-12]. However, research into the anticancer activity of rose petals has been limited. Here, we sought to investigate the effect of rose petal extract on the proliferation, migration, and invasion of lung and colon cancer cells.”
Reviewer 2 Report
Lim et al. evaluated the effect of rose petal extract (RPE) on the proliferation, migration, and invasion of A549, H1299, HCT116, and HT29 cancer cell lines. The authors concluded that RPE exhibited an inhibitory effect on cell proliferation, migration, and invasion which was associated with the inhibition of EGFR phosphorylation and subsequent downstream signaling.
Although the manuscript included some interesting results, some points require further clarification.
The manuscript may not be acceptable for publication until the authors perform major revisions.
Comments:
- The authors indicated that “PCNA expression was reduced with treatment of RPE”. If so, is DNA polymerase the target of RPE? Is it related to EGFR signaling? Please describe the possible mechanism of RPE.
- Figure 4A shows that while phosphorylation of EGFR was completely inhibited by 100 mg/ml of RPE, MEK1/2 activation was not inhibited. Therefore, MEK activation is not associated with EGFR activation. Accordingly, the interpretation of EGFR signaling seems arbitrary in Fig. 4A and 4C.
- In Figure 4, 10 mM of erlotinib seems a too high concentration. The results might be non-specific effects of EGFR inhibition.
- Is 100 mg/ml of RPE clinically relevant?
Author Response
Reviewer: 2
Comments to the Author
Lim et al. evaluated the effect of rose petal extract (RPE) on the proliferation, migration, and invasion of A549, H1299, HCT116, and HT29 cancer cell lines. The authors concluded that RPE exhibited an inhibitory effect on cell proliferation, migration, and invasion which was associated with the inhibition of EGFR phosphorylation and subsequent downstream signaling.
Although the manuscript included some interesting results, some points require further clarification. The manuscript may not be acceptable for publication until the authors perform major revisions.
Comment. The authors indicated that “PCNA expression was reduced with treatment of RPE”. If so, is DNA polymerase the target of RPE? Is it related to EGFR signaling? Please describe the possible mechanism of RPE.
Response. We appreciate for the reviewer’s detailed indication. We did not exclude the direct effect RPE on PCNA. However, we found RPE directly inhibit EGFR-related signaling pathway for anti-cancer activity. The expression of PCNA is used a biomarkers of proliferation marker of cancer cell as previous papers [6].
Comment. Figure 4A shows that while phosphorylation of EGFR was completely inhibited by 100 mg/ml of RPE, MEK1/2 activation was not inhibited. Therefore, MEK activation is not associated with EGFR activation. Accordingly, the interpretation of EGFR signaling seems arbitrary in Fig. 4A and 4C.
Response. We appreciate for the reviewer’s indication. Raf-MEK signaling pathway is not only modulated by EGFR. EGFR is a major upstream regulator of Raf-MEK signaling pathway. The first-line anti-cancer drugs (erlotinib, gefitinib, and afatinib) targets EGFR protein and these agents reveal suppressing activity of MEK signaling pathway concomitantly with EGFR suppression. Although we could not interpret the differences the reduction of between p-EGFR and p-MEK by RPE, we suggest the suppression of p-MEK is associated with EGFR suppression by RPE.
Comment. In Figure 4, 10 mM of erlotinib seems a too high concentration. The results might be non-specific effects of EGFR inhibition.
Response. We appreciate for the reviewer’s comment. Depending on the type of cell line, the concentration of erlotinib may be different. The typical concentration of erlotinib in the A549 cell line is thought to be 10 μM. Several references were indicated as shown in below [7,8].
Comment. Is 100 mg/ml of RPE clinically relevant?
Response. We appreciate for the reviewer’s comment. As per the reviewer’s indication, clinical study is required to verify the clinical relevance between100 mg/ml of RPE and the activity. We added the limits of current research to revised manuscript. However, oral administration of RPE 5 mg/kg B.W. revealed sufficient biological activity in mouse model (unpublished data, Manuscript preparation). Additionally, 10 mg/kg B.W. of RPE did not show any toxicity in mouse model.
Reference
- Liu, Q.; Yu, S.; Zhao, W.; Qin, S.; Chu, Q.; Wu, K. EGFR-TKIs resistance via EGFR-independent signaling pathways. Mol Cancer 2018, 17, 53, doi:10.1186/s12943-018-0793-1.
- Memmott, R.M.; Dennis, P.A. The role of the Akt/mTOR pathway in tobacco carcinogen-induced lung tumorigenesis. Clin Cancer Res 2010, 16, 4-10, doi:10.1158/1078-0432.CCR-09-0234.
- Jung, S.K.; Lee, M.H.; Lim, D.Y.; Lee, S.Y.; Jeong, C.H.; Kim, J.E.; Lim, T.G.; Chen, H.Y.; Bode, A.M.; Lee, H.J., et al. Butein, a Novel Dual Inhibitor of MET and EGFR, Overcomes Gefitinib-Resistant Lung Cancer Growth. Mol Carcinogen 2015, 54, 322-331, doi:10.1002/mc.22191.
- Bollu, L.R.; Katreddy, R.R.; Blessing, A.M.; Pham, N.; Zheng, B.; Wu, X.; Weihua, Z. Intracellular activation of EGFR by fatty acid synthase dependent palmitoylation. Oncotarget 2015, 6, 34992-35003, doi:10.18632/oncotarget.5252.
- Chakraborty, S.; Li, L.; Puliyappadamba, V.T.; Guo, G.; Hatanpaa, K.J.; Mickey, B.; Souza, R.F.; Vo, P.; Herz, J.; Chen, M.R., et al. Constitutive and ligand-induced EGFR signalling triggers distinct and mutually exclusive downstream signalling networks. Nat Commun 2014, 5, 5811, doi:10.1038/ncomms6811.
- Malkas, L.H.; Herbert, B.S.; Abdel-Aziz, W.; Dobrolecki, L.E.; Liu, Y.; Agarwal, B.; Hoelz, D.; Badve, S.; Schnaper, L.; Arnold, R.J., et al. A cancer-associated PCNA expressed in breast cancer has implications as a potential biomarker. Proc Natl Acad Sci U S A 2006, 103, 19472-19477, doi:10.1073/pnas.0604614103.
- Lelj-Garolla, B.; Kumano, M.; Beraldi, E.; Nappi, L.; Rocchi, P.; Ionescu, D.N.; Fazli, L.; Zoubeidi, A.; Gleave, M.E. Hsp27 Inhibition with OGX-427 Sensitizes Non-Small Cell Lung Cancer Cells to Erlotinib and Chemotherapy. Mol Cancer Ther 2015, 14, 1107-1116, doi:10.1158/1535-7163.MCT-14-0866.
- Simasi, J.; Schubert, A.; Oelkrug, C.; Gillissen, A.; Nieber, K. Primary and secondary resistance to tyrosine kinase inhibitors in lung cancer. Anticancer Res 2014, 34, 2841-2850.
Round 2
Reviewer 1 Report
There are still numerous questions have not been DIRECTLY addressed by the authors. For example, although, as responded, "previous studies, EGFR is regulate c-Raf-MEK-ERK and mTOR-Akt signaling pathway as a upstream modulator in carcinogenesis", the mechanisitc relationship between these two pathways, I personally believe, have not been investigated in the context of "RPE-treated lung cancer cells". The authors could have at least co-treated the cells with RPE and mTOR inhibitor, or RPE and EGFR inhibitor, and examined whether or not the two pathways would have been impacted.
Author Response
Reviewer: 1
Comment. There are still numerous questions have not been DIRECTLY addressed by the authors. For example, although, as responded, "previous studies, EGFR is regulate c-Raf-MEK-ERK and mTOR-Akt signaling pathway as a upstream modulator in carcinogenesis", the mechanisitc relationship between these two pathways, I personally believe, have not been investigated in the context of "RPE-treated lung cancer cells". The authors could have at least co-treated the cells with RPE and mTOR inhibitor, or RPE and EGFR inhibitor, and examined whether or not the two pathways would have been impacted.
Response. . We appreciate for the reviewer’s indication. The c-Raf-MEK-ERK and AKT-mTOR pathways presented in this study differ in detailed mechanisms involved in cell proliferation, but it is well known that both are downstream pathway of EGFR and are associated with cell proliferation. Reference papers related to this are presented as follows [1,2].
EGFR, which was more strongly inhibited by RPE, is already widely known to suppress the proliferation of A549 cells. In this study, A549 cell proliferation was inhibited not only by EGFR inhibitors but also by RPE. Of course, if these are co-treated, the proliferation inhibitory effect will be further increased, and in fact, the proliferation inhibition by co-treatment was confirmed in this study. On the other hand, RPE showed suppression of Raf and MEK belonging to the EGFR downstream, and down-regulation of AKT and mTOR by PTEN, an AKT inhibitor. Since it is well known that both Raf-MEK-ERK and AKT-mTOR pathways are governed by EGFR, it is presumed that RPE sufficiently reduced cell proliferation by regulating EGFR signaling pathways.
Reference
- Asati, V.; Mahapatra, D.K.; Bharti, S.K. PI3K/Akt/mTOR and Ras/Raf/MEK/ERK signaling pathways inhibitors as anticancer agents: Structural and pharmacological perspectives. Eur J Med Chem 2016, 109, 314-341, doi:10.1016/j.ejmech.2016.01.012.
- Bumrungthai, S.; Munjal, K.; Nandekar, S.; Cooper, K.; Ekalaksananan, T.; Pientong, C.; Evans, M.F. Epidermal growth factor receptor pathway mutation and expression profiles in cervical squamous cell carcinoma: therapeutic implications. J Transl Med 2015, 13, doi:ARTN 24410.1186/s12967-015-0611-0.
Reviewer 2 Report
A549 cells, which the authors used in this study, has KRAS mutation (G12S). Therefore, A549 cells are KRAS-driven cancer cell line, and then, this cell lines do not respond EGFR-TKI in a clinical relevant concentration, which is under 1 µM. This is an answer for my comment #2. I agree that the effect of rose petal extract (RPE) on the proliferation, migration, and invasion of A549, H1299, HCT116, and HT29 cancer cell lines. However, it might be not involved EGFR signaling. I will recommend the authors could change their title or use other kinds of cell lines.
Author Response
Reviewer: 2
Comment. A549 cells, which the authors used in this study, has KRAS mutation (G12S). Therefore, A549 cells are KRAS-driven cancer cell line, and then, this cell lines do not respond EGFR-TKI in a clinical relevant concentration, which is under 1 µM. This is an answer for my comment #2. I agree that the effect of rose petal extract (RPE) on the proliferation, migration, and invasion of A549, H1299, HCT116, and HT29 cancer cell lines. However, it might be not involved EGFR signaling. I will recommend the authors could change their title or use other kinds of cell lines.
Response. We appreciate for the reviewer’s comment. In your opinion, the A549 cell line is no a model with strong EGFR activation under normal conditions. However, RPE inhibited the basal phosphorylation level of this cell line. In addition, it is well known that the Raf-MEK-ERK and AKT/mTOR pathways inhibited by RPE are regulated by EGFR. In particular, as can be seen from the presented supplementary results, RPE is presumed to regulate the EGFR signaling pathway, considering the fact that the expression of cyclin D1 and c-myc increased by the treatment of EGF, a ligand of EGFR, was decreased by RPE. The EGFR-Raf-MEK-ERK signaling pathway is already well known as one of the major pathways involved in cell proliferation, migration and invasion. RPE showed EGFR inactivation, downregulation of EGFR downstream factors related to cell proliferation, and reduction proliferation of cancer cells excluding normal cells. These results suggest that RPE certainly influenced cell proliferation through regulation of the EGFR signaling pathway.